# The Effects of Indoor Pollutants Exposure on Allergy and Lung Inflammation: An Activation State of Neutrophils and Eosinophils in Sputum

**DOI:** 10.3390/ijerph17155413

**Published:** 2020-07-28

**Authors:** Khairul Nizam Mohd Isa, Zailina Hashim, Juliana Jalaludin, Leslie Thian Lung Than, Jamal Hisham Hashim

**Affiliations:** 1Department of Environmental and Occupational Health, Faculty of Medicine and Health Sciences, Universiti Putra Malaysia, UPM, Serdang 43400, Selangor, Malaysia; khairulnizamm@unikl.edu.my (K.N.M.I.); juliana@upm.edu.my (J.J.); 2Environmental Health Research Cluster (EHRc), Environmental Healthcare Section, Institute of Medical Science Technology, Universiti Kuala Lumpur, Kajang 43000, Selangor, Malaysia; 3Department of Microbiology and Parasitology, Faculty of Medicine and Health Sciences, Universiti Putra Malaysia, UPM, Serdang 43400, Selangor, Malaysia; leslie@upm.edu.my; 4IIGH United Nations University, UKM Medical Centre, Cheras 56000, Kuala Lumpur, Malaysia; jamalhas@hotmail.com

**Keywords:** lung inflammation, allergy, indoor pollutants, biomarkers, FeNO, eosinophil, neutrophil

## Abstract

Background: To explore the inflammation phenotypes following indoor pollutants exposure based on marker expression on eosinophils and neutrophils with the application of chemometric analysis approaches. Methods: A cross-sectional study was undertaken among secondary school students in eight suburban and urban schools in the district of Hulu Langat, Selangor, Malaysia. The survey was completed by 96 students at the age of 14 by using the International Study of Asthma and Allergies in Children (ISAAC) and European Community Respiratory Health Survey (ECRHS) questionnaires. The fractional exhaled nitric oxide (FeNO) was measured, and an allergic skin prick test and sputum induction were performed for all students. Induced sputum samples were analysed for the expression of CD11b, CD35, CD63, and CD66b on eosinophils and neutrophils by flow cytometry. The particulate matter (PM_2.5_ and PM_10_), NO_2_, CO_2_, and formaldehyde were measured inside the classrooms. Results: Chemometric and regression results have clustered the expression of CD63 with PM_2.5_, CD11b with NO_2_, CD66b with FeNO levels, and CO_2_ with eosinophils, with the prediction accuracy of the models being 71.88%, 76.04%, and 76.04%, respectively. Meanwhile, for neutrophils, the CD63 and CD66b clustering with PM_2.5_ and CD11b with FeNO levels showed a model prediction accuracy of 72.92% and 71.88%, respectively. Conclusion: The findings indicated that the exposure to PM_2.5_ and NO_2_ was likely associated with the degranulation of eosinophils and neutrophils, following the activation mechanisms that led to the inflammatory reactions.

## 1. Introduction

Exposure to indoor pollutants are strongly associated with increased morbidity and mortality, mainly among school children who spend most of their time in the classrooms [1]. Reports have shown that exposure to high concentration of particles, nitrogen dioxide (NO_2_), carbon dioxide (CO_2_), ozone (O_3_), volatile organic compounds (VOCs), and fibres induce persistent airway inflammation, which is mediated by the immune system [2,3]. Biomarkers, such as fractional exhaled nitric oxide (FeNO), cytokines, chemokines, lipid mediators, enzymes, adhesion molecules, and other growth factors, have been considered as the indicators of allergic airway inflammation [4]. The indicators that underlie the complex molecular pathways that regulate inflammation have not been fully elucidated; however, numerous studies have reported that the process certainly involves the activation of eosinophils and neutrophils [5,6]. They play roles in innate host defence via effector mechanisms, including degranulation, DNA traps, and cytolysis, following the activation cascaded by diverse mediators [7].

Extensive investigation of the mediators that are implicated in allergy, lung inflammation, and asthma has been documented but there was insufficient evidence available for the multi-dimensional characters of the activation and degranulation markers expression of CD11b, CD35, CD63, and CD66b on eosinophils and neutrophils. Activation markers, such as integrin Mac-1 (CD11b/CD18), L-selectin (CD62L), CBRM1/5, ICAM-1 (CD54), PD-L1 (CD274), PSGL-1 (CD162), FcγRII (CD32), CD16, CD44, and CD69, were shown to be upregulated on circulating or sputum granulocytes from asthma patients [8,9]. Response to allergens and environmental stimuli among asthma patients had a detrimental outcome and was characterized by the infiltration of different granule contents from eosinophils and neutrophils following activation [10]. Eosinophils and neutrophils have four distinct granules, namely azurophilic, secondary, tertiary, and secretary, which are formed throughout development in the bone marrow [11]. Several models have demonstrated the link between degranulation of these granules and pollutants exposure at different levels of stimuli with the expression of CD11c, CD63, and CD66b [12,13]. At present, biomarkers help tailor the management of respiratory illness and studies have highlighted precision therapy approaches based on disease mechanisms by targeting the cytokines and chemokines [14,15]. Recently, advanced statistical analysis approaches, such as hierarchical clustering and latent class analyses, have renewed the interest of researchers in the investigation of airway translational inflammation phenotypes [16,17]. This new vision helps the researchers to better understand and precisely describe the inflammatory phenotypes [18,19].

This study aimed to investigate the concentrations and sources of indoor pollutants in classrooms located in the urban and suburban areas of Hulu Langat, Selangor, Malaysia. Moreover, it aimed to predict the toxicodynamic effects of indoor pollutants in the classrooms towards marker expression on the eosinophils and neutrophils in sputum samples by using chemometric analysis techniques.

## 2. Materials and Methods 

### 2.1. Study Population

The study population was randomly sampled from eight secondary schools in Hulu Langat, Selangor, Malaysia. The researchers of this study targeted school children at the age of 14 of which they were randomly selected from four classrooms in each school. The total number of students who received their guardian’s consent and was thus recruited in the study was 470. Among them, only 50 (10.6%) students were diagnosed with asthma by a doctor based on the survey questionnaire. Another 46 students out of the remaining 420 students were randomly selected as a potential control group. The control group was selected among students who produced an adequate sputum cell count from the same class and school. In total, 96 students were included in the final study group. Students with a history of smoking in the last 12 months and students who received antibiotic treatments in the past four weeks were excluded from this study. The school areas were classified as urban and suburban by the Ministry of Education, Malaysia, based on the locale classification of the ecological measures. The data collection from the clinical assessments and indoor air monitoring were carried out at the same time in August until November 2018 and February 2019.

The researchers used the questionnaire adopted from the European Community Respiratory Health Survey (ECRHS) and International Study of Asthma and Allergies in Childhood (ISAAC) that are inclusive of questions on doctor-diagnosed asthma, allergies, and respiratory symptoms. The self-administered questionnaire was distributed to the selected students in the same week of the technical measurements. Subsequently, the researchers went through the questionnaires during a face-to-face interview with the students to clarify any uncertainty. The doctor-diagnosed asthma in this study is defined as having asthma medication, asthma attacks, and wheezes with breathlessness in the last 12 months, which were diagnosed by the physician [20,21]. This information was verified during face-to-face interviews and telephone calls with the students’ respective guardians.

### 2.2. FeNO Assessment and Allergy Skin Test

FeNO measurement was performed by a chemiluminescence analyser (NIOX VERO, Circassia, Sweden) as recommended by the American Thoracic Society/European Respiratory Society (ATS/ERS) [22]. The detection limits and accuracy for this device are 5–300 ppb and ±5 ppb, respectively. The students were asked to inhale deeply through the mouthpiece attached to the patient filter and then slowly exhale for about six to ten seconds at a constant flow rate (50 mL/s)—the single-breath technique. This process was repeated at least twice to get an average value. Students were instructed to refrain from eating and drinking for one hour before the FeNO assessment.

An allergy skin prick test was performed on the volar side of the forearm alongside *Dermatophagoides pteronyssinus* (Derp1) (house dust mite), *Dermatophagoides farina* (Derf1) (house dust mite), *Cladosporium herbarium* (fungi), *Alternaria alternate* (fungi), and *Felis domesticus* (cat) allergens in liquid form (Prick-Test Diagnostic, ALK-Abelló, Madrid, Spain). The same amount of histamine (10 mg/mL) and normal saline were used as the positive and negative controls, respectively. The reaction was measured after 15 min of which the wheal diameter was recorded. The allergen’s wheal diameter of 3 mm was considered as a positive control. Atopy was defined as a significant positive skin test reaction to at least one of the applied allergens [23].

### 2.3. Sputum Induction and Processing

Sputum induction was conducted by the inhalation of a nebulised, sterile mixture of 4.5% sodium chloride (hypertonic) and salbutamol 200 µg, followed by the coughing and expectoration of airway secretions. For nebulisation, an ultrasound nebuliser (Model CUN60 Citizen System Japan Co. Ltd., Tokyo, Japan) was was used as recommended [24] with a mouthpiece that fitted an output of ∼1 mL·min^−1^ to achieve successful sampling. The induced sputum samples collected from the respondents were kept in an icebox and further processed within two hours by using flow cytometry. The method of processing sputum samples has been previously described [25]. In short, the sputum sample was diluted with freshly prepared phosphate buffer saline and gently vortexed at room temperature for homogenisation. These steps were repeated thrice. Subsequently, the sputum samples were centrifuged at 800× *g* for 10 min. Next, cytospin slides of sputum cells were stained with May–Grunwald–Giemsa for the cell differentiation count. Samples with >80% of squamous epithelial cells were excluded for the flow cytometry analysis. 

### 2.4. Flow Cytometry

The processed sputum sample at a concentration of 1 × 10^6^ cells in 100 µL was resuspended in 1 mL of stain buffer (FBS) (BD Pharmingen^TM^, San Diego, CA, USA) and washed twice in cold stain buffer by centrifugation at 200× *g* for 5 min. Subsequently, the supernatant was removed, and the cell pellet was resuspended in 300 µL of stain buffer. A volume of 100 µL aliquots of the cell suspension were transferred into three different sterile polypropylene round-bottom tubes, and monoclonal antibodies and isotype controls were added to the cells according to the manufacturer protocol. The cells were immunostained with antibodies for neutrophil and eosinophil surface markers of PE-Cy 7 Mouse Anti-Human CD11b/Mac-1, BB515 Mouse Anti-Human CD35, PE Mouse Anti-Human CD63, APC-H7 Mouse Anti-Human CD16, PerCP-Cy 5.5 CD41a, and Alexa Fluor 647 Mouse Anti-Human CD66b purchased from BD Biosciences, US. The isotype controls of PE-Cy 7 Mouse IgG1, BB515 Mouse IgG1 K, PE Mouse IgG1 K, APC-H7 Mouse IgG1 K, PerCP-Cy 5.5 Mouse IgG1 K, and Alexa Fluor 647 Mouse IgM K (BD Biosciences, US) were added to the cells. The cell samples were incubated for 15 min at room temperature in the dark. Next, the cell samples were washed twice in 1 mL of stain buffer and centrifuged at 300× *g* for 5 min. The supernatant was removed, and the cells were loosened up by tapping the tube. Subsequently, the cells were carefully resuspended in 500 µL of the stain buffer. Samples without antibodies and isotype were used as the controls. Hereafter, the cells were analysed by using the BD FACSCanto II flow cytometry instrument (BD Bioscience, US) equipped with blue, red, and violet lasers. Compensation was set to account for spectral overlap between the four fluorescent channels. The gating region was set so that less than 1% of the samples were stained with negative controls. Data were processed by using FlowJo software (Version 10.1r1) [26,27].

The activation of eosinophils was measured by using CD11b (integrin αM), CD35 (CR1), and CD66b (CEACAM-8). Meanwhile, CD11b was used as an activation marker for neutrophils. The degranulation activity was measured by the expression of CD11b and CD63 for eosinophils as a marker for cyctalloid (specific/secondary) and secretory (sombrero) vesicles, respectively. For neutrophils, the CD11b, CD35, CD63, and CD66b markers were used as tertiary, secretory, azurophilic, and specific granule expression markers in the degranulation activities [28]. Isotype controls were used as the positive control and to address the background produced by non-specific antibody binding, whereas the Anti-Mouse Ig K Negative Control Compensation Particle Set was used to optimise the fluorescence compensation settings [29] (Figure 1). 

### 2.5. Assessment of Indoor Air Quality in Classrooms and Building Inspection

Indoor pollutants and physical parameters, including temperature (°C), relative humidity (%), and carbon dioxide (ppm), were measured in the classrooms during learning session within an hour by using a Q-TrakTM IAQ monitor (Model 7565 TSI Incorporated, Shoreview, MN, USA) with the average log interval values over one minute. The accuracy of this device on temperature, relative humidity, and CO_2_ are ±0.6 °C, ±3%, and ±50 ppm, respectively. The sampling for the particles was measured by using a Dust-Trak monitor (Model 8532 TSI Incorporated, Shoreview, MN, USA) at a sampling rate of 1.7 L/min with a resolution of 0.001 mg/m^3^ and detection limit of 0.001–150 mg/m^3^. The PPM Formaldemeter^TM^ htV-M (PPM Technology Ltd, Wales, UK) with accuracy of 10% at 2 ppm was used to measure the concentration of formaldehyde. In each school, a total of four hours of measurements were taken from four randomly selected classrooms for a period of an hour each during the learning session, as has been previously described [30,31,32]. The instruments used were placed one metre from the ground in the centre of the classrooms. All instruments used were calibrated regularly. The NO_2_ (µg/m^3^) concentration was measured by using a diffusion sampler (IVL, Goteborg, Sweden) for a period of a week. The sampler was place at height of approximately 2–3 m above the ground and returned to the IVL Swedish Environmental Research Institute Laboratory (Goteborg, Sweden), an accredited laboratory for further analysis. This measurement technique provides an average concentration of NO_2_ in the air during a week, with a limit of detection (LOD) of 0.5 µg/m^3^ and a 10% (at the 95% confidence level) measurement uncertainty [33]. A building inspection was carried out before the indoor air quality assessments were obtained. Details on the building information, floor furnish, furniture, and type of ventilation system were noted [34].

### 2.6. Ethical Statement

The Ethics Committee for Research Involving Human Subjects Universiti Putra Malaysia (JKEUPM) has approved this study (JKEUPM-2018-189) and each of the students was given a written consent form for their guardian’s approval.

### 2.7. Data Analysis

The descriptive analysis was carried out by Mann–Whitney tests using the Statistical Package for the Social Sciences (SPSS) 25.0. The differences in biomarker expression between the doctor-diagnosed asthmatic children and healthy children were made by using GraphPad Prism 8 for Windows. Subsequently, a principle component analysis (PCA) and agglomerative hierarchical clustering (AHC) were applied to explore the association and pattern recognition between the biomarker expression and the concentrations of indoor pollutants. The final prediction models were generated by logistic regression analysis in which the models’ performance was based on the coefficient of determinant, overall accuracy, sensitivity, specificity, and the area under the curve (AUC) of the receiver operator characteristics (ROC) [35]. The researchers used the standardized data of the indoor pollutants and biomarkers in the chemometric and regression analyses. The multivariate analysis was carried out by using the Statistical Package XLSTAT Evaluation 2019.2.3 (Addinsoft, New York, US).

## 3. Results

### 3.1. Data Analysis for the Personal and Clinical Characteristics of School Children

The study population was well-balanced between the doctor-diagnosed asthmatic children (52%) and healthy children (48%), in which 60% of them were from urban schools. The majority of asthmatic children tested positive for at least one of the allergens, with 74.0% of them sensitised towards house dust mites (Derp1 and Derf1), followed by cat dander (30.0%) (Appendix A). The FeNO levels were statistically higher among the doctor-diagnosed asthmatic children than healthy children (*p* < 0.001). The researchers observed that the total percentage of eosinophil count in the sputum samples was slightly higher in the doctor-diagnosed asthmatic children but not statistically different between the two groups (*p* > 0.05) (Table 1).

Both the eosinophils and neutrophils expressed an activated phenotype in both groups of induced sputum samples. The expression of CD11b was significantly upregulated in both their cell surfaces among the doctor-diagnosed asthmatic children (*p* < 0.05). The expression of CD63 on the neutrophils was also significantly higher in the sputum samples of the doctor-diagnosed asthmatic children as compared to healthy children (*p* < 0.001). Samples from the doctor-diagnosed asthmatic children with a high expression of CD11b (tertiary) and CD63 (azurophilic/crystalloid) on the eosinophil and neutrophil surfaces indicated a moderately degranulated state of sputum granulocytes (Figure 2).

### 3.2. Levels of Indoor Pollutants and Building Inspection Data

All the classrooms were designed with natural ventilation, which is equipped with glass windowpanes on both sides of the wall. The classrooms were equipped with an average of three ceiling fans in each classroom. The schools were painted, and the floor surface was furnished with concrete. There were bookshelves, whiteboard, and soft boards in every classroom. Some of the classrooms had window curtains fixed on both sides of the class. The statistical analysis showed a significant difference in all of the indoor environmental parameters between the urban and suburban schools (Table 2).

### 3.3. Chemometrics Analysis of the Biomarkers and Indoor Air Pollutants

A PCA was performed to explain the variance observed between the biomarkers and indoor pollutants in a more efficient way. Prior to the PCA, Bartlett’s sphericity and Kaiser–Meyer–Olkin (KMO) tests were conducted to determine the correlation difference and sampling adequacy, with both measurements achieving the required levels. The PCA was applied with a normalization procedure and the coefficient factor loadings produced also expressed the correlation between the variables [36]. The factor loadings of the four factors with eigenvalues >1 were extracted from the eosinophil and neutrophil expression markers. A total of 39.0% of the variation for both Factor 1 and Factor 2 was observed in the expression of markers on eosinophils tested with indoor pollutants. Moderate factor loadings were identified for expression of CD11b and CD35, together with a strong factor loading for the concentrations of NO_2_ and formaldehyde in Factor 1. In other words, the upregulation of CD11b and CD35 expression in the eosinophils were associated with exposure to NO_2_ and formaldehyde; in turn, Factor 2 had moderate factor loadings of FeNO levels, PM_10_, and PM_2.5_. Moderate factor loadings were observed for FeNO levels, CD66b, and PM_10_ in Factor 3, with a total variation of 14.9%. High factor loading of CD63 was observed in Factor 4, with a total variation of 13.0%. For neutrophils, a strong factor loading of CD66b expression together with moderate factor loadings for FeNO levels, CD11b, and CD63 were recorded in Factor 1, with a variation of 24.2%. Factor 2 showed a strong factor loading for the concentration of formaldehyde together with a moderate factor loading for the concentrations of CO_2_, PM_10_, and PM_2.5_, with 19.3% of the total variation. There were moderate loading factors for expression of CD35 and the concentrations of NO_2_ and PM_10_ in Factor 3, with 14.3% of the total variation. Factor 4 showed moderate factor loadings of CD11b expression and concentration of PM_2.5_, with 13.2% of the total variation (Table 3).

We further analysed the biomarkers and indoor pollutants to categorize them based on their homogeneity levels using agglomerative hierarchical clustering (AHC). Our data showed the cluster of markers and indoor pollutants from the PCA and AHC analyses were relatively identical. Three clusters were generated for eosinophils, which consisted of CD66b, FeNO levels, and concentrations of CO_2_ in Cluster 1, with 98.4% of variance within-class; whereas Cluster 2 presented 56.5% of the variation within-class for CD11b expression and concentrations of NO_2_ and formaldehyde. High (82.9%) variance within-class was observed for CD35 and CD63 expression and concentrations of PM_10_ and PM_2.5_ in Cluster 3; three clusters were also generated for neutrophils, which consisted of CD35, CD63, CD66b, and concentrations of CO_2_, PM_10_, and PM_2.5_, with 95.7% of the variation within-class for Cluster 1. The concentrations of NO_2_ and formaldehyde were grouped in Cluster 2 with 26.0% of the variation within-class. Cluster 3 showed 56.7% of the variation within-class for CD11b expression and FeNO levels. The clusters generated through this process confirmed the contributing factors in the PCA earlier. This showed that exposure to CO_2_, PM_10_, and PM_2.5_ have resulted in degranulation of both eosinophils and neutrophils, with upregulation of the markers for tertiary (CD11b), specific (CD66b), and azurophilic/crystalloid (CD63). In this study, exposure to NO_2_ and formaldehyde also triggered the activation of tertiary eosinophil surface markers (Figure 3A).

### 3.4. Binary Logistic Regression (LR)

Finally, models were built to predict the toxicodynamic effects of indoor pollutants towards marker expression on the eosinophil and neutrophil in the sputum samples among doctor diagnosed asthmatic children. For this objective, the researchers next used the binary logistic regression to model the prediction with the potential confounders of gender, atopy, parental asthma/allergy status, and area of schools. Cluster 1 showed an overall accuracy of 76.0% in predicting asthmatic children by using CD66b expression markers on the eosinophil and FeNO levels in relation to CO_2_ exposure. Meanwhile, the model generated for Cluster 2 showed a 76.0% accuracy and 68.0% sensitivity in predicting asthmatic children by using the upregulation of CD11b expression on eosinophils in relation to the NO_2_ exposure. In the model generated for Cluster 3, the upregulation of CD63 expression on eosinophils and PM_2.5_ concentration was significantly associated (*p* < 0.05), with a 71.9% accuracy and 60.0% sensitivity. Similarly, the upregulation of the neutrophil expression markers, CD63 and CD66b, was significant (*p* < 0.05), and could be predicted from the PM_2.5_ exposure with a 72.9% accuracy and 72.0% sensitivity. Overall, children with a status of atopy, parental asthma/allergy, and from urban school were more likely to develop asthma (*p* < 0.05) (Table 4 and Figure 4).

## 4. Discussion

The role of biomarkers in airways is complex and specific, which is helpful in evaluating the aetiology, characterisation of phenotyping, and treatment of allergy and lung inflammation [37]. In this study, the FeNO levels were significantly higher among asthmatic school children, which are similar with the studies conducted in China [38], Terengganu, Malaysia [39], and Penang, Malaysia [40]. The result showed that there was inflammation in the airways and the average value was above the threshold of 50 ppb, which could reflect a high degree of inflammation. Liu et al. [41] and Carlsen et al. [42] reported that there was a significantly positive relationship between the FeNO levels and almost all pollutants, namely PM_10_, PM_2.5_, SO_2_, NO_2_, CO, and VOCs. This advocates a relationship between the high levels of all pollutants measured inside the classroom of urban schools and the high levels of FeNO among school children in this study. Some researchers estimated that FeNO is positively correlated up to five-fold and two-fold when exposed to NO_2_ [43] and finer particles, such as PM_2.5_ [44], respectively, which could be modulated by DNA methylation in the arginase–nitric oxide synthase pathway [45,46].

The CO_2_ concentration in both school areas was below the recommended limit of 1000 ppm [47]. Similarly, the PM_10_ and PM_2.5_ concentrations were below the 24 h mean of the World Health Organisation (WHO) guideline (PM_10_ = 50 µg/m^3^, PM_2.5_ = 25 µg/m^3^), the National Ambient Air Quality Standard by USEPA (PM_10_ = 150 µg/m^3^, PM_2.5_ = 35 µg/m^3^), and the new Malaysian Ambient Air Quality Standard 2018 Interim Target-2 (PM_10_ = 120 µg/m^3^, PM_2.5_ = 50 µg/m^3^) [48]. A few classrooms recorded a concentration of PM_2.5_ that exceeded the value of 25 µg/m^3^, especially in the urban areas (37.5%) compared to the suburban (12.5%) areas. The median level of NO_2_ for the urban and suburban areas was also below the WHO guideline of 40 µg/m^3^ (annual mean), with only 18.8% and 25.0% of the classrooms in the urban and suburban areas, respectively, exceeding the limit. Overall, the levels of indoor air pollutants were below the guideline limits. This was due to the sufficient natural ventilation system and a wider window design on both sides of the classroom, together with the adequately equipped ceiling fans. The classroom design has a well-balanced ventilation that suits the temperature of the equatorial region and is able to reduce the particles, NO_2_, and CO_2_ concentrations [49]. Additionally, Silvestre et al. [50] reported that an opening of 56% of the classroom windows under natural ventilation conditions was able to keep the CO_2_ concentration below 1000 ppm.

The schools, classrooms, and children were randomly selected from all secondary schools in the Hulu Langat area, Malaysia. Thus, we concluded that this study was not seriously influenced by selection bias. Moreover, Malaysia has a similar climate all year around; therefore, with the natural ventilation flowing through the windows in the classrooms, the indoor and outdoor levels of pollutants would be expected to be constant throughout the year. This is supported by several studies that have determined equal indoor to outdoor (I/O) ratios for PM_10_, PM_2.5_, NO_2_, CO, and VOCs measured in schools across Peninsular Malaysia [51,52,53]. Be that as it may, the cross-sectional study design utilized here preludes making conclusions on causality.

In contrast to earlier findings, the total percentage of eosinophil and neutrophil counts in the sputum sample was not statistically different between the doctor-diagnosed asthmatic children and healthy children groups. Previous studies reported that the percentage of eosinophils and neutrophils for asthmatic children was significantly different and in the range of 2.5–13.0% and 15–47%, respectively [54,55,56]. Meanwhile, for healthy subjects, the percentages were in the range of 0.5–4.0% and 24.1–37.0%, respectively [57,58].

There have been few recent studies on activation and degranulation marker expression in the sputum samples of asthmatic children. The finding of this study confirmed that the sputum granulocytes of the asthmatic children increased the expression of the classical activation markers, CD11 and CD63, in both eosinophil and neutrophil cells, as reported by Tak et al. [28]. The upregulation of these tertiary and azurophilic/crystalloid granules is associated with the circulating cytokines that occur sequentially in response to the stimulus [59]. The mitogen-activated protein kinase (MAPK) pathway is believed to be central to the degranulation process [60]. The present study failed to show the upregulation of CD35 expression on eosinophils. In accordance with the study by Berends et al. [61], the downregulation of CD35 in the sputum of asthmatic children could be partly explained by the absence of intracellular stores for CD35 on eosinophils and neutrophils. Another possible explanation for this discrepancy was that CD35 is highly expressed on blood eosinophils or circulating granulocytes and was only directly associated with antigen inhalation [62] or the lower threshold stimulus required for cell activation [63].

The CD66b (CEACAM8) is a single-chain GPI-anchored glycoprotein and was recognised as an exclusive degranulation marker for neutrophils [64]. CD66b is upregulated when neutrophils are activated. The researchers of this study observed that the expression of CD66b was slightly upregulated on the surface of eosinophils and neutrophils collected from the airways of asthmatic children as compared to healthy children. It represents a normal activation pattern of neutrophils in relation to the migration from the circulating blood in vessels [65]. The late-phase response of the neutrophils could also possibly increase the CD11b, CD11b/18, CD35, CD64, and CD66b expressions [66]. The researchers found one study that identified that CD11b, CD16, and CD66b were consistently expressed on the neutrophils surface and were independent of their location and level of activation [67]. This could be contributed by the increased levels of intracellular cyclic GMP that yielded upregulation in the CD63 and CD66b expression on neutrophils [68].

PCA and AHC are very helpful approaches for dimensionality reduction in proteomics data. In this current study, these analytical approaches depicted similar group factors of biomarkers and air pollutants. The final regression analysis generated relatively moderate prediction models. The researchers noted that the upregulation of CD63 expression on both leukocytes and CD66b expression on neutrophils was related to particle exposure. The likely risks were observed among children under atopic and parental asthmatic/allergic conditions and children from schools located in urban areas. This finding reinforces the previous in vitro study conducted by Jin et al. [69]. They suggested that particulate allergens potentiated the mast cells to modulate the recruitment of eosinophils in the airways by internalising the particulate allergens into the CD63^+^ intracellular compartments through an endolytic pathway. Another in vitro study reported that CD66b only activated the neutrophils in the peptidoglycan challenge but did not upregulate the surface activation of eosinophils [70]. This is the possible explanation of why CD66b expression is clustered and associated with PM_10_ and PM_2.5_ in neutrophils but not with eosinophils in the AHC and regression analyses. Likewise, the study conducted by Banerje et al. [71] using flow cytometry analysis reported that there were increased CD35, CD16, and CD11b/CD18 expression on circulating neutrophils and a high percentage of eosinophils in the sputum of adults who have been exposed to PM_10_ and PM_2.5_.

To the researchers’ knowledge, this study was the first study that explored the interrelation of CD11b, CD63, CD35, and CD66b marker expression on eosinophils and neutrophils with different parameters of air pollutants. This study revealed that CD11b was not clustered together with PM_10_ and PM_2.5_. This result was coherent with the study conducted by Ishii et al. [72] using immunocytochemistry, which showed that the expression of CD11b on alveolar macrophages was unaffected after two hours of stimulation with PM_10_. They suggested that the adhesive interaction between CD11/CD18 on alveolar macrophages with CD54 on the bronchial epithelial cells contributed to the amplification of cytokine production from the alveolar macrophages. The chemometrics analysis in this current study was also clustered and showed a significant relationship between the CD11b expression on eosinophils with NO_2_, especially among asthmatic children under atopic and parental asthmatic/allergic conditions. This finding was consistent with the review article by Hiraiwa and Eeden [73] and an in vitro study reported by Hodgkins et al. [13]. They found that dendritic cells expressed an upregulation of CD11b at 48 h during NO_2_-promoted allergic sensitisation. A study has shown that CD11b is directly involved in cellular adhesion, which is expressed in many leukocytes, including neutrophil, monocytes, natural killer cells, and macrophages. The migration of these leukocytes to the inflammation site will only take place if the CD18 subunit is present [74]. The other possible roles of CD11b were reported by Medoff et al. [75] in their experiment in which the CD11b^+^ had critical roles in mediating the Th2 cell and eosinophil recruitment in the airways via STAT6-dependent chemokine production.

This study also showed that the FeNO levels were positively correlated with the expression of the activation (CD66b, CD11b) and degranulation (CD66b, CD11b) markers for both leukocytes. This result is consistent with the results in the previous studies conducted by Guo et al. [76] and Kobayashi et al. [77], who also indicated that FeNO levels were reflected by eosinophilic airway inflammation [78]. In fact, the activated neutrophils can recruit the Th17/IL-17 and Th1 cells via chemokine release [79] and cause neutrophil infiltration within the airways [80]. This finding also reinforces that eosinophils and eosinophils are the binary indicators for the phenotyping of asthma. It was found in previous studies that exposure to PM_10_ was significantly associated with the increased levels of FeNO in healthy children tested on robust multi-pollutant models [41,81,82,83]. In line with this report, the chemometric analysis results in this study provided further evidence on the positive effects of PM_10_ on bronchial inflammation and resulted in the increase of FeNO levels among children.

The cluster approach used in this study, which is aimed at improving the interpretability of the data, interestingly revealed that the formaldehyde and NO_2_ concentrations were in the same factor, with a total variance of 26.0%. This finding confirmed that the formation of formaldehyde was through the photochemical reactivity of NO_2_ in the air with VOCs generating different aldehydes [84]. Formaldehyde also originates from furniture made out of wood and plastics, plywood, textile, table laminate, and consumer products, which is commonly available in the classroom [85]. As reported by Hua [86], NO_2_ potentially originates from the process of fossil-fuel combustion, biomass burning, and agricultural activities. Hereafter, all the schools in this current study are located very close to the main road and industrial area, which was considered the probable sources of NO_2_ in the classrooms. The researchers found that PM_10_ and PM_2.5_ were grouped together in the dimensionality analysis of the PCA and AHC, and this indicates that both particles originated from the same source. The primary sources of particles in the urban and suburban areas were industrial emissions, transportation, and traffic emissions [87,88]. It was possible that the indoor particles also originated from the occupant’s activities or re-suspension of deposited particles, soil materials from the school children’s shoes, skin flakes, furniture fragments, and less frequent cleaning [89]. Children are at risk of day-long exposure to the same indoor pollutants, not only at school, but also at home—which was reflected in the time spent during non-school days. This possibility merits further investigation.

## 5. Conclusions

In conclusion, the chemometric analysis methods produced robust and immunologically meaningful results, which clustered the degranulation markers (CD11b and CD63) expressed on eosinophils with the concentration of NO_2_ and PM_2.5_. Besides that, the degranulation markers expressed on the neutrophils, CD63 and CD66b, were clustered together with the PM_2.5_ concentration. A further prospective study is now obligatory to validate the models generated from this current study.

## Figures and Tables

**Figure 1 ijerph-17-05413-f001:**
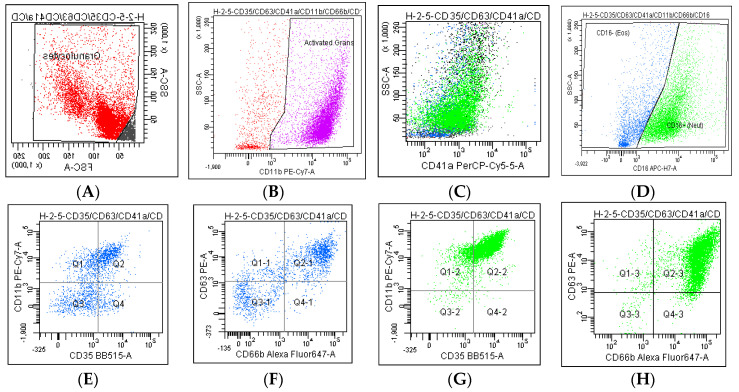
Gating strategy to identify activated sputum granulocytes. (**A**) A forward-/side-scatter (FSC/SSC) gate to identify the granulocytes. (**B**) The activated granulocytes were based on CD11b+. (**C**) CD41a was gated with SSC to confirm that the activated granulocytes were neutrophils and eosinophils. (**D**) Subsequently, the eosinophils (blue) and neutrophils (green) were separated based on the expression of CD16− and CD16+ on SSC, respectively. The activation markers of CD11b, CD35, CD63, and CD66b for eosinophils (**E**,**F**) and neutrophils (**G**,**H**) were based on the negative gates of the isotype control for all antibodies.

**Figure 2 ijerph-17-05413-f002:**
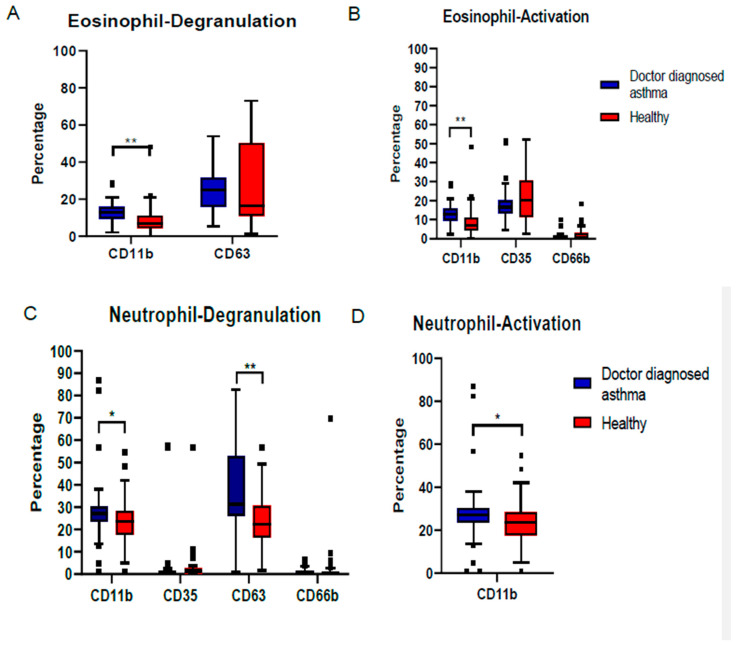
Expression profile of sputum granulocytes compared between the doctor-diagnosed asthmatic and healthy school children. Expression of degranulation and activation markers on eosinophil (**A**,**B**) and neutrophil (**C**,**D**). The expression of CD11b as the degranulation and classical activation marker is displayed twice in this graph. * *p* < 0.05, ** *p* < 0.001.

**Figure 3 ijerph-17-05413-f003:**
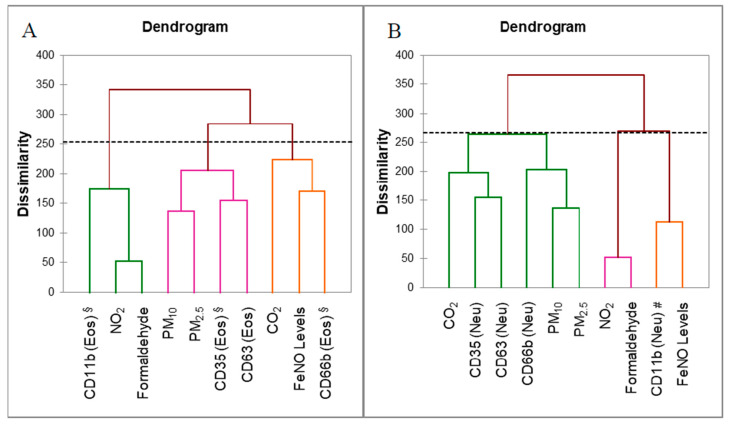
Agglomerative hierarchical clustering (AHC) analysis using the Ward linkage method and using Euclidean distances to generate the clustering of degranulation and the activation of markers and environmental pollutants measured inside classrooms for (**A**) eosinophils and (**B**) neutrophils. The dotted line represents the pruning level to generate distinct clusters. § Activation marker for eosinophils; # Activation marker for neutrophils.

**Figure 4 ijerph-17-05413-f004:**
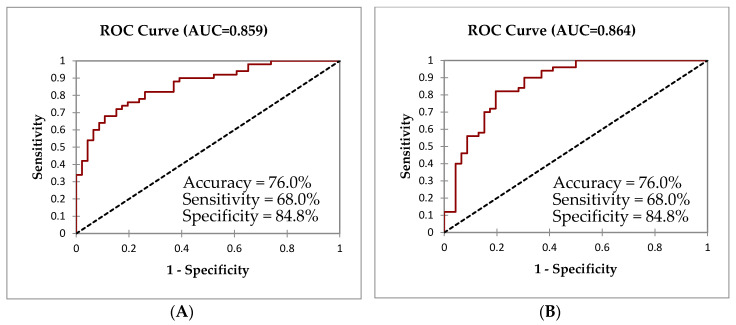
Receiver operating curve (ROC) for the models predicting an asthmatic or healthy child classification based on the clusters generated in the AHC analysis. The corresponding predictors for each curve are presented in Table 4. For eosinophils: (**A**) ROC for Cluster 1, (**B**) Cluster 2, and (**C**) for Cluster 3. For neutrophils: (**D**) ROC for Cluster 1 and (**E**) for Cluster 3.

**Table 1 ijerph-17-05413-t001:** Students’ characteristic and inflammation status for the doctor-diagnosed asthma and control group.

Characteristics	Doctor Diagnosed Asthma (*n* = 50)	Healthy (*n* = 46)	*p*-Value
Female (%)	44.2	55.8	0.094
Male (%)	61.4	38.6	
Atopic (%)	65.0	35.0	<0.001 **
Non atopic (%)	30.6	69.4	
Parental with asthma/allergy (%)	65.9	34.1	0.020 *
Parental without asthma/allergy (%)	41.8	58.2	
School area—urban (%)	60.4	39.6	0.102
School area—suburban (%)	43.8	56.3	
Clinical characteristics			
FeNO levels (ppb)	56 (66.5)	23 (32.3)	0.002 *
Eosinophil count (%)	11.6 (11.3)	10.2 (9.2)	0.259
Neutrophil count (%)	11.6 (5.0)	13.4 (14.0)	0.130

Values are the median (IQR) for clinical characteristics. IQR = Interquartile range. * *p* < 0.05; ** *p* < 0.001.

**Table 2 ijerph-17-05413-t002:** Comparison of the environmental parameters between schools located in urban and suburban areas.

Parameter	Urban *n* = 16			Suburban *n* = 16			*p*-Value
	Median (IQR)	Min	Max	Median (IQR)	Min	Max	
Temperature (°C)	29.0 (2.0)	28.0	32.0	27.5 (1.0)	27.0	28.0	<0.001 **
Relative Humidity (%)	74.7 (9.5)	63.6	88.1	81.4 (7.5)	74.8	88.1	<0.001 **
Formaldehyde (mg/m^3^)	13.2 (9.3)	5.2	19.5	3.1 (5.2)	2.1	15.9	<0.001 **
CO_2_ (ppm)	453.0 (34.5)	417.0	468.0	455.5 (25.5)	402.0	471.0	0.462
NO_2_ (µg/m^3^)	32.0 (7.0)	15.0	45.0	19.0 (22.5)	16.0	48.0	<0.001 **
PM_2.5_ (µg/m^3^)	24.6 (2.4)	17.9	26.5	22.0 (1.9)	16.8	26.9	<0.001 **
PM_10_ (µg/m^3^)	41.6 (7.5)	34.7	48.0	37.0 (4.9)	32.3	44.9	<0.001 **

*n* = 32; IQR = Interquartile range, Min = Minimum, Max = Maximum. ** *p* < 0.001.

**Table 3 ijerph-17-05413-t003:** Factor loadings using PCA for eosinophils and eosinophils. The moderate (0.5–0.75) and strong (>0.75) factor loadings are highlighted in bold.

Biomarkers and Indoor Pollutants	Eosinophils				Neutrophils			
	F1	F2	F3	F4	F1	F2	F3	F4
FeNO levels	0.194	**−0.575**	**0.501**	0.235	**0.653**	−0.306	−0.108	−0.120
CD11b	**0.528**	0.083	−0.398	0.059	**0.571**	−0.053	0.175	**0.685**
CD35	**0.559**	0.129	0.432	0.348	0.337	0.213	**0.612**	−0.445
CD63	−0.193	0.379	0.070	**0.732**	**−0.665**	0.394	0.117	−0.288
CD66b	0.285	−0.465	**0.561**	−0.419	**0.727**	0.067	−0.235	−0.039
CO_2_	−0.346	−0.329	−0.456	0.084	−0.287	**−0.506**	0.123	−0.360
NO_2_	**0.771**	−0.116	−0.118	0.375	0.470	0.301	**0.657**	−0.412
PM_10_	0.102	**0.582**	**0.584**	−0.035	0.074	**0.684**	**−0.523**	−0.047
PM_2.5_	−0.053	**0.699**	−0.061	−0.490	−0.409	**0.503**	−0.099	**0.502**
Formaldehyde	**0.788**	0.380	0.100	0.123	0.322	**0.746**	0.467	−0.089
Eigenvalue	2.10	1.81	1.49	1.30	2.42	1.93	1.43	1.32
Variability (%)	21.0	18.1	14.9	13.0	24.2	19.3	14.3	13.2
Cumulative %	21.0	39.0	54.0	67.0	24.2	43.4	57.8	71.0

**Table 4 ijerph-17-05413-t004:** Summary of the binary logistic regression models based on the clusters generated in the AHC analysis.

Variable	Β	SE	*p*-Value	R^2^
***Eosinophils***				
**Cluster 1**				
Constant	−0.860	0.409	0.035 *	0.494
FeNO Levels	0.615	0.222	0.006 *	
CD66b	−0.850	0.250	<0.001 **	
CO_2_	0.457	0.191	0.016 *	
Atopy	0.893	0.377	0.018 *	
Parental Asthma/Allergy	0.643	0.319	0.044 *	
Area—Urban	0.747	0.354	0.035 *	
Cluster 2				
Constant	−0.109	0.536	0.840	0.504
CD11b	0.454	0.260	0.018 *	
NO_2_	1.305	0.375	0.002 *	
Formaldehyde	−0.684	0.440	0.120	
Atopy	0.993	0.378	0.009 *	
Parental Asthma/Allergy	0.916	0.424	0.031 *	
**Cluster 3**				
Constant	−1.387	0.462	0.003 *	0.377
CD35	−0.097	0.189	0.608	
CD63	−0.080	0.166	0.042 *	
PM_10_	−0.379	0.212	0.074	
PM_2.5_	−0.378	0.181	0.037 *	
Atopy	0.909	0.377	0.015 *	
Parental Asthma/Allergy	0.842	0.350	0.016 *	
***Neutrophils***				
**Cluster 1**				
Constant	−1.039	0.490	0.034 *	0.510
CD35	−0.054	0.169	0.751	
CD63	−0.632	0.256	0.014 *	
CD66b	−1.794	0.858	0.036 *	
CO_2_	0.230	0.200	0.250	
PM_10_	−0.253	0.218	0.244	
PM_2.5_	−0.582	0.234	0.013 *	
Atopy	1.134	0.439	0.010 *	
Parental Asthma/Allergy	0.944	0.404	0.021 *	
Area—Urban	1.348	0.514	0.009 *	
**Cluster 3**				
Constant	−0.876	0.432	0.042 *	0.298
FeNO Levels	0.063	0.171	0.712	
CD11b	0.390	0.197	0.047 *	

* *p* < 0.05, ** *p* < 0.001.

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
