# Peer review of "The Effects of Indoor Pollutants Exposure on Allergy and Lung Inflammation: An Activation State of Neutrophils and Eosinophils in Sputum"

_ijerph, 2020, doi:10.3390/ijerph17155413_

Round 1

Reviewer 1 Report

The paper “the effects of indoor pollutants exposure on allergy and lung inflammation: an activation state of neutrophils and eosinophils in sputum” studied the human exposure responses to indoor pollutants such as particulate matter, NO2, and CO2. The authors focused on allergy and lung inflammation and used the inflammation biomarker FeNO as suggested in previous studies. Four expression markers of CD11b, CD35, CD63 and CD66b on eosinophil and neutrophil were also measured and compared between groups with asthma and healthy ones. I have a few concerns.

L24-25
Please do not use acronyms here.

L29
Switch the word “impressive”.

L31
What model is used here? Logistic regression?

Introduction
The intro is lack of depth and current understanding of the studied subject. Please add more content here.

L95
Please list the detection limit and uncertainty of FeNO measurements.

L143
Please add a reference for the software.

L168
The authors should specify the dust-trak is used for PM measurements. And list its detection limit and uncertainty/accuracy.

L205 “the expression of CD63 on neutrophil was also significant … healthy children (p<0.001)”
Figure 2(C) suggests the opposite trend. Please verify the statement.

L223
Please list the specification of the passive samplers.

L281 Figure 3
Please explain why CO2 is added to the model. References 2-3 do not suggest CO2 contributes to allergy and lung inflammation.
The model does not include other pollutants such as VOCs, SO2, and CO. Please discuss any limitations in the paper.

L314
Please explain the meaning of a negative constant in the regression model.

L340
Need to indicate the USEPA standards are 24 hours based.

Author Response

Thank you for your constructive comments on our manuscript. We have addressed your comments one by one below and revised our manuscript based on your suggestions.

L24-25
Please do not use acronyms here.

Response 1: Full name for acronyms were included in this line. “International Study of Asthma and Allergies in Children (ISAAC) and European Community Respiratory Health Survey (ECRHS)” 

L29 
Switch the word “impressive”.

Response 2: This word was deleted.

L31 
What model is used here? Logistic regression?

Response 3: Referring to the first sentence of the Result section:” Chemometric and regression results have clustered the expression of CD63 with PM2.5,…”

Introduction
The intro is lack of depth and current understanding of the studied subject. Please add more content here.

Response 4: More discussion was added in the Introduction part.

L95 
Please list the detection limit and uncertainty of FeNO measurements.

Response 5: This information was added in Section 2.2 “The detection limits and accuracy for this device are 5-300 ppb and ± 5 ppb, respectively.”

L143 
Please add a reference for the software.

Response 6: References were included for this software. “Data were processed by using FlowJo software (Version 10.1r1) [26],[27].”

L168
The authors should specify the dust-trak is used for PM measurements. And list its detection limit and uncertainty/accuracy.

Response 7: Information on the detection limit and uncertainty/accuracy was included for all instruments used in this study.

“… The accuracy of this device on temperature, relative humidity, and CO2 are ± 0.6°C, ± 3%, and ± 50 ppm, respectively. The sampling for the particles was measured by using Dust-Trak monitor (Model 8532 TSI Incorporated, USA) at a sampling rate of 1.7 L/min with a resolution of 0.001 mg/m3 and detection limit of 0.001-150 mg/m3. Formaldehyde TV-M Meter (PPM Technology Ltd, UK) with accuracy of 10% at 2 ppm was used to measure the concentration of formaldehyde…. This measurement technique provides an average concentration of NO2 in the air for during a week with the limit of detection (LOD) of 0.5µg/m3 and 10% (at 95% confidence level) of measurement uncertainty [32].”

L205 “the expression of CD63 on neutrophil was also significant … healthy children (p<0.001)”
Figure 2(C) suggests the opposite trend. Please verify the statement.

Response 8: Thanks. The explanation was correct as the researcher referred back to raw and analysed data. The correction was done in the graph C (Figure 2).

L223 
Please list the specification of the passive samplers.

Response 9: Details explanation on passive sampler was specified in the Section 2.5.

“The NO2 (µg/m3) concentration was measured by using diffusion sampler (IVL, Sweden) for a period of a week. The sampler was place at height of approximately 2-3 metres above the ground and returned to IVL Swedish Environmental Research Institute Laboratory (Goteborg, Sweden), an accredited laboratory for further analysis. This measurement technique provides an average concentration of NO2 in the air for during a week with the limit of detection (LOD) of 0.5µg/m3 and 10% (at 95% confidence level) of measurement uncertainty [32].”

L281 Figure 3
Please explain why CO2 is added to the model. References 2-3 do not suggest CO2 contributes to allergy and lung inflammation.
The model does not include other pollutants such as VOCs, SO2, and CO.

Response 10: The CO2 was included in the model as referred to the articles below. Studies reported that CO2 was associated with high prevalence of wheeze in children. 

https://link.springer.com/article/10.1007/s00420-015-1076-4

  1.  

Please discuss any limitations in the paper.

Response 11: More discussion on study limitation was added in Discussion part.

“The schools, classrooms and children were randomly selected from all secondary schools in Hulu Langat area, Malaysia. Thus, we concluded that this study was not seriously influenced by selection bias. Moreover, Malaysia has a similar climate all year around, therefore with the natural ventilation flow through the windows in the classrooms, indoor and outdoor levels of pollutants would be expected to be constant throughout the year. This is supported by several studies that have determined equal indoor to outdoor (I/O) ratios for PM10, PM2.5, NO2, CO, and VOCs measured in schools across Peninsular Malaysia [50]–[52].  Be that as it may, the cross-sectional study design utilized here preludes making conclusions on causality."

L314 
Please explain the meaning of a negative constant in the regression model.

Response 12: The negative values for Constant was aligned with the transformed (standardize (n)) data used in this analysis. Both biomarkers and environmental parameters data were transformed and used in the chemometric and regression analyses.

L340 
Need to indicate the USEPA standards are 24 hours based.

Response 13: The WHO guideline, USEPA and Malaysian Standard was referred to 24 hours mean values and added in the sentence.

“Similarly, the PM10 and PM2.5 concentrations were below the 24 hours mean  of World Health Organisation (WHO) guideline (PM10= 50 µg/m3, PM2.5= 25 µg/m3 ),.. ”

Reviewer 2 Report

There are severe methodological errors in the study presented:
1. Contamination by PM2.5 and PM10 particulate matter suffers cross effects, mainly from external air pollutants from mobile and fixed sources.
2. The feed may affect the quantitative of trace elements in the methodology of analysis of these elements, chosen by the authors.
3. 3. The aetiology of contamination suffers other cross-effects originating from the place where the students reside, circulate and, above all, the indoor contamination of their homes.

Author Response

Thank you for your constructive comments on our manuscript. We have addressed your comments one by one below and revised our manuscript based on your suggestions.

Contamination by PM2.5 and PM10 particulate matter suffers cross effects, mainly from external air pollutants from mobile and fixed sources.

Response 1: Natural ventilation design applies in most of the schools in tropical countries become major health issues, especially at urban and sub-urban areas. The pollutants may originate from various sources and were discussed in the last part of the Discussion.

The feed may affect the quantitative of trace elements in the methodology of analysis of these elements, chosen by the authors.

Response 2: Good suggestion. This merit further study in the future with detail out the specific carcinogenic trace elements exposure, especially in the dust samples.

The aetiology of contamination suffers other cross-effects originating from the place where the students reside, circulate and, above all, the indoor contamination of their homes.

Response 3: Researchers added a discussion on this part 

Round 2

Reviewer 1 Report

The authors answered my questions.

Reviewer 2 Report

The changes made after the review by colleagues and the justifications given by the authors made me change my mind. Therefore I suggest the publication of the article—success to the authors.